# Experimental realization of entanglement in multiple degrees of freedom between two quantum memories

Wei Zhang[1,2], Dong-Sheng Ding[1,2], Ming-Xin Dong[1,2], Shuai Shi[1,2], Kai Wang[1,2], Shi-Long Liu[1,2], Yan Li[1,2], Zhi-Yuan Zhou[1,2], Bao-Sen Shi[1,2] & Guang-Can Guo[1,2]

Entanglement in multiple degrees of freedom has many benefits over entanglement in a single one. The former enables quantum communication with higher channel capacity and more efficient quantum information processing and is compatible with diverse quantum networks. Establishing multi-degree-of-freedom entangled memories is not only vital for high-capacity quantum communication and computing, but also promising for enhanced violations of nonlocality in quantum systems. However, there have been yet no reports of the experimental realization of multi-degree-of-freedom entangled memories. Here we experimentally established hyper- and hybrid entanglement in multiple degrees of freedom, including path (K-vector) and orbital angular momentum, between two separated atomic ensembles by using quantum storage. The results are promising for achieving quantum communication and computing with many degrees of freedom.

[1] Key Laboratory of Quantum Information, CAS, University of Science and Technology of China, Hefei, Anhui 230026, China. [2] Synergetic Innovation Center of Quantum Information & Quantum Physics, University of Science and Technology of China, Hefei, Anhui 230026, China. Correspondence and requests for materials should be addressed to D.-S.D. (email: dds@ustc.edu.cn) or to B.-S.S. (email: drshi@ustc.edu.cn).

Quantum entanglement has a vital role in various quantum information fields, including quantum communication[1], quantum computation[2], teleportation[3] and quantum cryptography[4]. Acting as information carriers, photons can be entangled not only in a single degree of freedom (DOF), such as in polarization[5], spatial-mode[6], time-bin[7] and path[8] entanglement, but also in multiple DOFs independently, as in hyperentanglement[9,10], or in multiple DOFs mutually, as in hybrid entanglement[11]. Entanglement in multiple DOFs offers many advantages over entanglement in a single DOF; for example, multi-DOF hyperentanglement enables more efficient Bell measurements[12–14] and makes superdense coding that breaks the conventional linear-optics threshold achievable[15,16]. Meanwhile, entanglement in multiple DOFs can be utilized in asymmetrical optical quantum networks[17] and to generate multi-qubit entangled states[18]. Furthermore, hyperentanglement has many applications in improving techniques for state purification[19], achieving quantum computing[15,20] and testing nonlocality in quantum mechanics[21].

Alternatively, entanglement in multiple DOFs can exploit the advantages of different DOFs; for example, photons entangled in the polarization or time-bin DOF can be efficiently transmitted through an optical fibre, whereas photons encoded in orbital angular momentum (OAM) space offer improved channel capacity in the fields of both classical[22] and quantum[23,24] information. Multi-DOF entanglement that includes the OAM DOF has many superior properties; for example, it can increase the information carried by a single photon[25], further enhance the channel capacity and improve the efficiency of a network[22], and close the detection loophole in Bell experiments[26].

From this point of view, entanglement in multiple DOFs is very promising for quantum networks[27] in terms of enhancing channel capacity and improving compatibility. Such a quantum network requires quantum storage to establish entanglement between different nodes. Over the past decade, the quantum storage of entanglement in single DOFs has been achieved in many different quantum memory systems[28–33]; however, the storage of entanglement in multiple DOFs is still a challenge because of the difficulty of simultaneously achieving coherent control of multiple DOFs. Recently, the quantum storage of $2 \otimes 2$ hyperentanglement in the polarization and time-bin DOFs in a solid memory was reported[34]; in that study, light-memory hyperentanglement was established using the atomic frequency comb technique. However, memory–memory entanglement in multiple DOFs, which would represent a critical step towards a multi-DOF quantum network, has not yet been reported.

Here we report on the realization of the quantum storage of entanglement in multiple DOFs, including hyperentanglement and hybrid entanglement, based on cold atomic ensembles and the consequent establishment and verification of memory–memory entanglement. First, $2 \otimes 3$ hyperentanglement, consisting of two-dimensional (2D) entanglement between a collective atomic excited state (also called a spin wave) and photonic polarization as well as three-dimensional (3D) OAM entanglement between a spin wave and a single photon, is established between one atomic ensemble and a single photon through spontaneous Raman scattering (SRS). This entanglement is generated through an innovative method based on constructing a phase-insensitive interferometer, which allows the system to generate any of the four Bell states and operate for a long period of time without any locking technique. Then, this single photon is sent to and stored in another atomic ensemble, thereby creating spin-wave entanglement between the two atomic ensembles and thus successfully establishing memory–memory

hyperentanglement. Finally, we retrieve both spin waves to single photons and evaluate their entanglement. We construct the density matrices for the photon–photon 2D entanglement in the polarization DOF with a fidelity of $89.7 \pm 3.8\%$ and the photon–photon 3D entanglement in the OAM DOF with a fidelity of $91.1 \pm 4.5\%$. We also demonstrate the violation of the Clauser-Horne-Shimony-Holt (CHSH) inequality with no noise correction. Furthermore, we demonstrate hybrid entanglement in the path and OAM DOFs between two ensembles with only a small change in the experimental set-up. Our experimental results demonstrate the successful creation of memory-memory entanglement in multiple DOFs.

## Results

**Establishing hyperentanglement in multiple DOFs**. The medium used here to generate entanglement in multiple DOFs is an optically thick ensemble of $^{85}$Rb atoms trapped in a 2D magneto-optical trap (MOT)[35]. A simplified illustration of the experimental set-up is presented in Fig. 1. A single photon (Signal 1) with a 795-nm wavelength hyperentangled with the spin wave in MOT A is created with the aid of a beam displacer after illumination with light (Pump 1). Because of the conservation of momentum in the SRS process, the initial system has zero linear momentum and OAM. Thus, the resulting joint state of Signal 1 and the spin wave has zero momentum in both K-vector space and OAM space, so the spin wave in MOT A is entangled with the Signal 1 photon. The beam displacer here is used to coherently superpose the state of Signal 1 from a different K-vector space to the same path. This hyperentangled state is expressed as shown in equation 1 (hereafter, we use the term 'path' to represent the corresponding K-vector throughout the text). The generated Signal 1 photon is then delivered to the second atomic ensemble in MOT B for storage. Beam displacer 3 and beam displacer 4, which form an interferometer, are used to guarantee the same memory efficiency for differently polarized states of the Signal 1 photon. When we shut off the coupling light, the Signal 1 photon is stored in MOT B as an atomic spin wave, thus establishing hyperentanglement between the spin waves of the two atomic ensembles. In this case, the light-memory hyperentanglement is converted into memory–memory hyperentanglement. After 100 ns of storage of the spin wave in MOT B and 200 ns of storage of the spin wave in MOT A, we retrieve both spin waves to single photons by turning on the coupling light and the Pump 2 light, respectively. Before being collected, both the Signal 1 and Signal 2 (retrieved from the spin wave in MOT A) photons are sent to two analysers: a polarization analyser consisting of a polarizing beam splitter and wave plates, and a spatial mode analyser (spatial light modulator (SLM), HOLOEYE: LETO). Afterwards, both single photons are filtered through cavities (not depicted in Fig. 1) and conducted to single-photon detectors (avalanche diodes, Perkin-Elmer SPCM-AQR-15-FC) through fibres. We reconstruct the density matrices of the photon-photon states before and after the storage of Signal 1 and check the CHSH inequality to describe the entangled states in the polarization DOF and the OAM DOF. When we exchange the regions surrounded by dotted lines as depicted in Fig. 1b and simultaneously block the vertical memory path $M$ in MOT B, hybrid entanglement in the path and OAM DOFs between the two atomic ensembles can be experimentally established.

Here hyperentanglement, consisting of path-polarization entanglement between a spin wave and a single photon as well as OAM entanglement between a spin wave and a single photon, is directly generated by illumination with Pump 1 through the SRS process in MOT A. The generated state (unnormalized) can

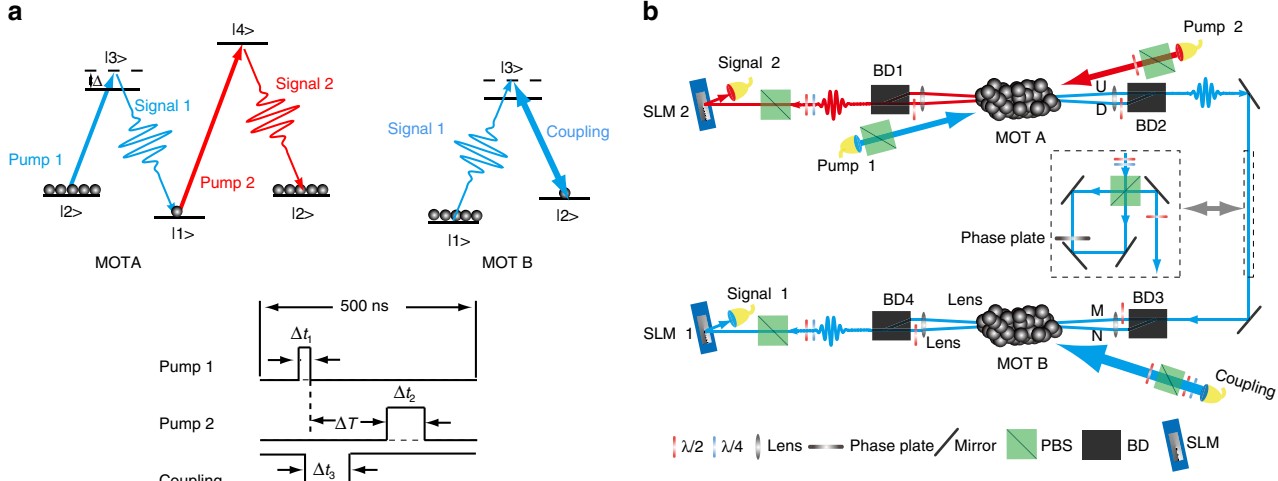

**Figure 1 | Generation and storage of entanglement in multiple DOFs.** (**a**) Energy diagram and time sequence. Pump 1 and Pump 2 are pulses with durations of $\Delta t_1 = 30$ ns and $\Delta t_2 = 200$ ns, respectively. The storage time set by $\Delta T$ for the spin wave in MOT A is 200 ns, and the storage time set by $\Delta t_3$ for the spin wave in MOT B is 100 ns. $\Delta$, which represents single-photon detuning, is set to $+70$ MHz, $|1\rangle = |5S_{1/2}, F = 2\rangle$, $|2\rangle = |5S_{1/2}, F = 3\rangle$, $|3\rangle = |5P_{1/2}, F = 3\rangle$, $|4\rangle = |5P_{3/2}, F = 3\rangle$. (**b**) Simplified experimental set-up. PBS, polarizing beam splitter; $\lambda/2$, half-wave plate; $\lambda/4$, quarter-wave plate; BD, beam displacer; SLM, spatial light modulator; $U$ ($D/M/N$), represents the path. Pump 1 is obliquely incident on the atomic ensemble with the same angle (1.5°) with respect to path $U$ and path $D$ in MOT A, and Pump 2 counter-propagates with Pump 1 through the atomic ensemble. The coupling light is also obliquely incident on the atomic ensemble in MOT B with the same angle of 1.5° with respect to path $M$ and path $N$. The set-up depicted in the region enclosed by the dotted rectangle on the left is used for preparing hybrid entanglement. The powers of the Pump 1, Pump 2 and coupling light beams are 0.1, 4 and 24 mW, respectively.

be expressed as

$$|\psi_1\rangle = \left(|D_A\rangle|H_{S1}\rangle + e^{i\theta_1}|U_A\rangle|V_{S1}\rangle\right)$$
$$\otimes \left(\sum_{m=-1}^{m=1} c_m|-m_A\rangle|m_{S1}\rangle\right) \tag{1}$$

where $|D_A\rangle$ and $|U_A\rangle$ refer to the spin waves related to paths $U$ and $D$, respectively, in MOT A; $|H_{S1}\rangle$ and $|V_{S1}\rangle$ represent the generated horizontal and vertical polarizations, respectively, of the Signal 1 photon; and $\theta_1$ is the phase difference between paths $U$ and $D$, which is set to zero in this experiment. $|-m_A\rangle$ represents the OAM eigenmode of the atomic spin wave in MOT A, with quanta of $-m$, and $|m_{S1}\rangle$ is the OAM eigenmode of the Signal 1 photon, with quanta of $m$. $|c_m|^2$ is the excitation probability. Here the value of $m$ ranges from $-1$ to 1. Usually, $c_m = c_{-m}$.

With the aid of the Mach–Zehnder interferometer formed by beam displacer 3 and beam displacer 4 and two half-wave plates, the generated Signal 1 photon that is hyperentangled with the spin wave in MOT A is stored in MOT B. Thus, hyperentanglement between the two atomic ensembles is established as follows:

$$|\psi_2\rangle = \left(|D_A\rangle|N_B\rangle + |U_A\rangle|M_B\rangle\right)$$
$$\otimes \left(|R_A\rangle|L_B\rangle + \alpha|G_A\rangle|G_B\rangle + |L_A\rangle|R_B\rangle\right) \tag{2}$$

where $|N_B\rangle$ and $|M_B\rangle$ refer to the spin waves related to paths $N$ and $M$, respectively, in MOT B; $\alpha = c_{m=0}/c_{m=1}$; and $c_{m=1} = c_{m=-1}$. Hereafter, we use $|L\rangle$, $|G\rangle$, and $|R\rangle$ to represent the states $|-1\rangle$, $|0\rangle$, and $|1\rangle$, respectively. After 100 ns of storage of Signal 1 as an atomic spin wave in MOT B and 200 ns of storage of the spin wave in MOT A, we turn on the coupling light and the Pump 2 light to retrieve both spin waves as single photons. The resulting

photon–photon state can be expressed as

$$|\psi_2'\rangle = \left(|H_{S2}\rangle|H_{S1}\rangle + |V_{S2}\rangle|V_{S1}\rangle\right)$$
$$\otimes \left(|L_{S2}\rangle|L_{S1}\rangle + \alpha|G_{S2}\rangle|G_{S1}\rangle + |R_{S2}\rangle|R_{S1}\rangle\right) \tag{3}$$

where $|\psi_x'\rangle$ represents the retrieved photon–photon state corresponding to the entangled state $|\psi_x\rangle$. Experimentally, the two states $|\psi_1\rangle$ and $|\psi_2\rangle$ are characterized by retrieving them as the photon–photon states $|\psi_1'\rangle$ and $|\psi_2'\rangle$ for further analysis.

We first reconstruct the density matrices of the photon–photon entanglement in the polarization DOF and the OAM DOF using the standard method[36] (see Methods). For the 2D polarization entanglement, the real parts of the density matrices for the photon–photon states before ($|\psi_{1(P)}'\rangle$) and after ($|\psi_{2(P)}'\rangle$) storage are shown in Fig. 2a,b. For the 3D OAM entanglement, the real parts of the constructed density matrices before ($|\psi_{1(OAM)}'\rangle$) and after ($|\psi_{2(OAM)}'\rangle$) storage are shown in Fig. 2c,d. These related states are expressed as follows:

$$\left|\psi_{1(P)}\right\rangle = |D_A\rangle|H_{S1}\rangle + |U_A\rangle|V_{S1}\rangle$$
$$\left|\psi_{2(P)}\right\rangle = |D_A\rangle|N_B\rangle + |U_A\rangle|M_B\rangle$$
$$\left|\psi_{2(P)}'\right\rangle = |H_{S2}\rangle|H_{S1}\rangle + |V_{S2}\rangle|V_{S1}\rangle$$
$$\left|\psi_{1(OAM)}\right\rangle = |R_A\rangle|L_{S1}\rangle + \alpha|G_A\rangle|G_{S1}\rangle + |L_A\rangle|R_{S1}\rangle$$
$$\left|\psi_{2(OAM)}\right\rangle = |R_A\rangle|L_B\rangle + \alpha|G_A\rangle|G_B\rangle + |L_A\rangle|R_B\rangle$$
$$\left|\psi_{2(OAM)}'\right\rangle = |L_{S2}\rangle|L_{S1}\rangle + \alpha|G_{S2}\rangle|G_{S1}\rangle + |R_{S2}\rangle|R_{S1}\rangle$$
$$\tag{4}$$

For the 2D polarization entanglement, the fidelity (see Supplementary Note 3) of the polarization-entangled state before storage compared with the ideal state is $87.7 \pm 2.4\%$, and the fidelity, which quantifies how closely the state after storage resembles the state before storage, is $89.7 \pm 3.8\%$. For the 3D OAM entanglement,

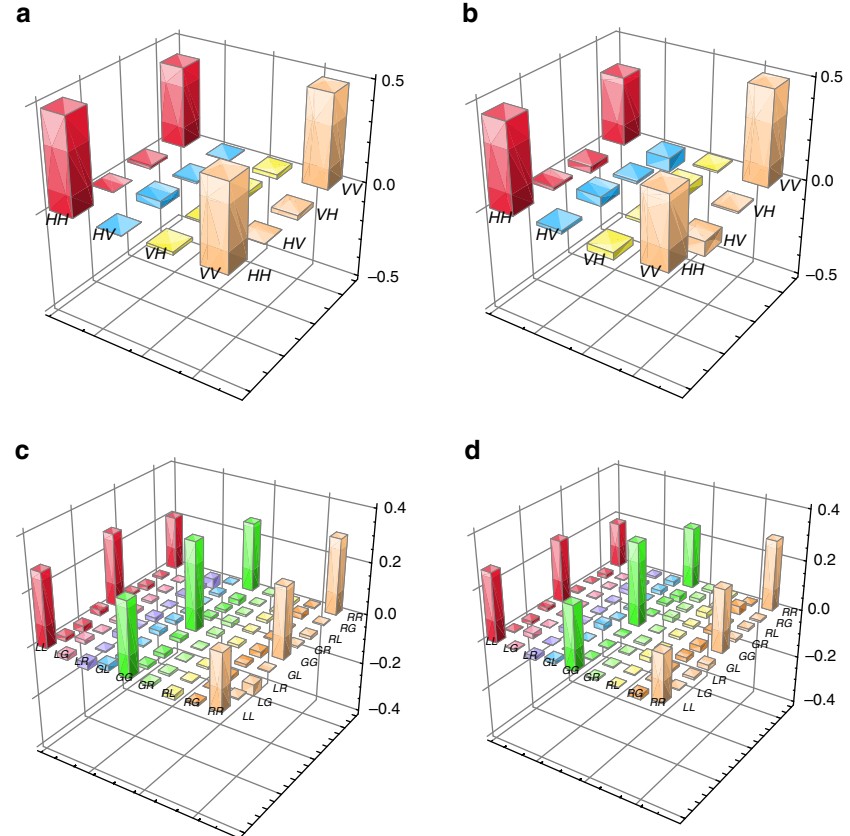

**Figure 2 | Density matrices for hyperentanglement.** The real parts of the constructed density matrices for the 2D polarization entanglement (**a,b**) and the 3D OAM entanglement (**c,d**), before (**a,c**) and after (**b,d**) storage. The values of the imaginary parts of the constructed density matrices are very low; they are distributed in the ranges of 0 ± 4.8% before storage and 0 ± 4.4% after storage for the 2D polarization entanglement and within the ranges of 0 ± 3.5% before storage and 0 ± 5.3% after storage for the 3D OAM entanglement.

the corresponding fidelities are $79.7 ± 2.7\%$ and $91.1 ± 4.5\%$, respectively, where the low fidelity value of $79.7 ± 2.7\%$ is due to the existence of $\alpha$, which is equal to $\sim 1.1$ in our experiment.

Second, we demonstrate the entanglement before and after storage by checking whether the CHSH Bell inequality is violated. For the polarization entanglement, we measure the correlation function $E(\theta_2, \theta_1)$, which can be calculated from the rate of the coincidence of differently polarized states, with $\theta_1(\theta_2)$ being the polarization angle for the Signal 1 (Signal 2) photon. We obtain the CHSH parameter $S_P = |E(\theta_2, \theta_1) - E(\theta_2, \theta'_1) + E(\theta'_2, \theta_1) + E(\theta'_2, \theta'_1)|$, with $\theta_2 = 0$, $\theta_1 = \pi/8$, $\theta'_2 = \pi/4$ and $\theta'_1 = 3\pi/8$. Here, the $S_P$ values obtained are $2.60 ± 0.03$ before storage and $2.51 ± 0.05$ after $\sim 100$ ns of storage time without any noise correction. For the 3D OAM entanglement, we check the CHSH inequality in the subspace (listed below) by varying the phase angles of SLM 1 and SLM 2:

$$\left| \psi_{1(\text{OAM-2D})} \right\rangle = \left| R_A \right\rangle \left| L_{S1} \right\rangle + \left| L_A \right\rangle \left| R_{S1} \right\rangle$$

$$\left| \psi_{2(\text{OAM-2D})} \right\rangle = \left| R_A \right\rangle \left| L_B \right\rangle + \left| L_A \right\rangle \left| R_B \right\rangle \qquad (5)$$

$$\left| \psi'_{2(\text{OAM-2D})} \right\rangle = \left| L_{S2} \right\rangle \left| L_{S1} \right\rangle + \left| R_{S2} \right\rangle \left| R_{S1} \right\rangle$$

Using the same method used in ref. 31, the calculated $S_{\text{OAM-2D}}$ values are $2.47 ± 0.05$ before storage and $2.32 ± 0.08$ after storage without noise correction.

In Fig. 3, we show interference patterns for entanglement. The average visibility of interference for the polarization entanglement

is 92.4% before storage and 88.4% after storage. For the 2D OAM entanglement, the average visibility is 84.0% before storage and 77.5% after storage. All visibility values are larger than the threshold of 70.7%, which clearly proves that entanglement is created and preserved in our quantum memory system.

It is obvious from equation (1) that the entanglements in the two DOFs are independent of each other. Therefore, a sufficient demonstration of hyperentanglement requires completely independent measurements for every DOF[34] or joint measurements including multiple DOFs[10]. However, because of the low count rate of photons in cold atomic media (see Supplementary Note 4) and the different memory efficiencies between different DOFs, the ability to employ joint measurements including multiple DOFs is limited because this method requires quite a long measurement time and is subject to low fidelity. Therefore, we adopt the method of measuring entanglement in every DOF independently. However, our SLMs are applicable to horizontally polarized light only. Two SLMs are shutdown when we measure the entanglement between the spin wave and photonic polarization before and after storage; thus, this is an OAM-independent measurement. However, when we measure the data to construct the density matrices of the 3D OAM entanglement or to check the Bell inequality for the 2D OAM entanglement in the subspace, we must ensure that the photons passing through the SLMs are horizontally polarized, as depicted in Fig. 1b. In this case, the measurement of OAM entanglement is NOT independent of polarization. For completeness, we check the polarization entanglements in the subspace of the OAM DOF to verify the

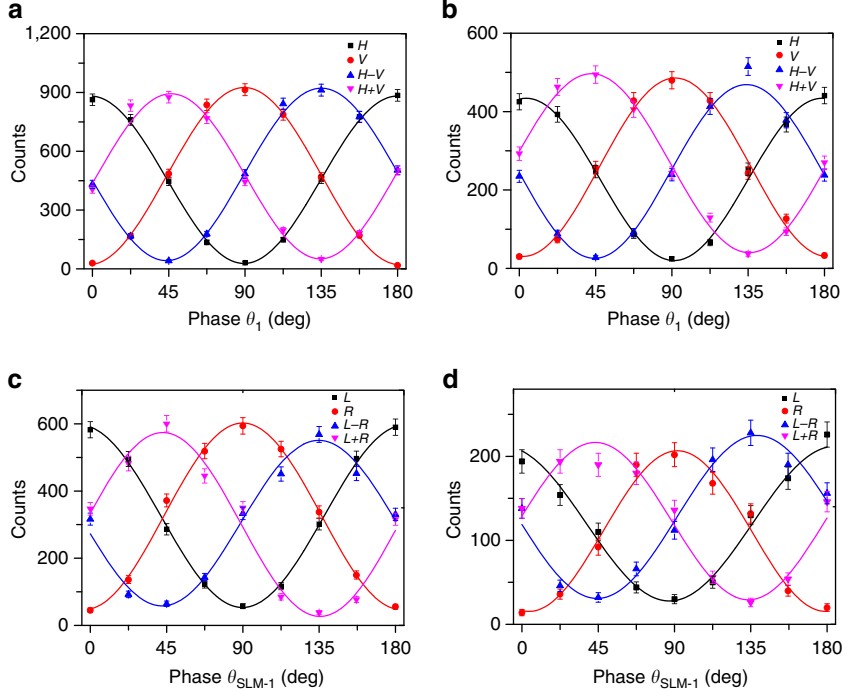

**Figure 3 | Interference curves for entanglement.** The interference curves before (**a,c**) and after (**b,d**) storage, for the 2D polarization (**a,b**) and 2D OAM (**c,d**) entanglements. The curves in **a,b** represent the coincidence rates for different values of $\theta_1$ when the Signal 2 photon is projected into the states $|H\rangle$, $|V\rangle$, $(|H\rangle - |V\rangle)^{1/2}$, and $(|H\rangle + |V\rangle)^{1/2}$. The curves in **c,d** represent the coincidence rates for different values of $\theta_{SLM-1}$ when the Signal 2 photon is projected into the states $|L\rangle$, $|R\rangle$, $(|L\rangle - R\rangle)^{1/2}$ and $(|L\rangle + |R\rangle)^{1/2}$. The error bars are estimated from Poisson statistics and represent ± s.d.

independence (see Supplementary Figs 1,2, Supplementary Table 1 and Supplementary Note 1).

**Establishing hybrid entanglement in multiple DOFs.** Another important type of state that takes advantage of multiple DOFs is hybrid entanglement. Whereas hyperentanglement is the entangling of states in multiple DOFs independently, hybrid entanglement is the entangling of states in multiple DOFs mutually, which has a similar form to that discussed in ref. 17. With a small change in the experimental set-up, we can generate polarization-path entanglement between the Signal 1 photon and the spin wave in MOT A by utilizing the principle of momentum conservation in K-vector space. Then, using a suitably designed Sagnac interferometer, we convert the photonic polarization information into OAM information (see Supplementary Fig. 3 and Supplementary Note 2). In this way, we establish hybrid entanglement between the photon in the OAM DOF and the spin wave in the path DOF; this state is expressed as $\left|\psi_{1\text{-hybrid}}\right\rangle$. We input this photon encoded in the OAM DOF into the atomic ensemble in MOT B for storage as a spin wave, thereby establishing hybrid entanglement between the two spin waves in the two separated atomic ensembles, which is expressed as $\left|\psi_{2\text{-hybrid}}\right\rangle$.

To characterize the nature of the hybrid entanglement between the two separated atomic ensembles, we map the state of the spin waves in these memories onto the photonic state ($\left|\psi'_{2\text{-hybrid}}\right\rangle$) by turning on the Pump 2 and coupling pulse fields after 100 ns of storage in MOT B and 200 ns of storage in MOT A. For entanglement without storage, we turn on only the Pump 2 light to convert this entanglement into photon–photon entanglement ($\left|\psi'_{1\text{-hybrid}}\right\rangle$), while blocking the coupling light and the ensemble in MOT B. We obtain density matrices for $\left|\psi'_{1\text{-hybrid}}\right\rangle$ and $\left|\psi'_{2\text{-hybrid}}\right\rangle$ through projection measurements (see the Methods),

with the results shown (only the real parts) in Fig. 4a,b. The related states are expressed as follows:

$$\left|\psi_{1\text{-hybrid}}\right\rangle = |D_A\rangle|L_{S1}\rangle + |U_A\rangle|R_{S1}\rangle$$
$$\left|\psi_{2\text{-hybrid}}\right\rangle = |D_A\rangle|L_B\rangle + |U_A\rangle|R_B\rangle \qquad (6)$$
$$\left|\psi'_{2\text{-hybrid}}\right\rangle = |H_{S2}\rangle|L_{S1}\rangle + |V_{S2}\rangle|R_{S1}\rangle$$

We calculate the fidelity of this entanglement by comparing $\left|\psi'_{1\text{-hybrid}}\right\rangle$ with an ideal maximum entangled state, thereby obtaining a value of $94.6 \pm 1.4\%$ before storage. After storage, we perform quantum tomography for $\left|\psi'_{2\text{-hybrid}}\right\rangle$ and obtain the density matrix shown in Fig. 4b. By comparing the density matrices of $\left|\psi_{2\text{-hybrid}}\right\rangle$ and $\left|\psi'_{2\text{-hybrid}}\right\rangle$, we calculate the fidelity of hybrid entanglement to be $93.6 \pm 1.4\%$.

At the same time, we measure the two-photon interference to characterize the hybrid entanglement property. The coincidence rates of the two photons are measured in different bases for Signal 1 and Signal 2. In the bases of $(|H\rangle - |V\rangle)/2^{1/2}/(|H\rangle - i|V\rangle)/2^{1/2}$ for Signal 2, we change the relative phase between $|R\rangle$ and $|L\rangle$ for Signal 1 and obtain the interference curves shown in Fig. 5a,b, which correspond to the interference before and after storage, respectively. The average visibility is 93.1% before storage and 84.6% after storage. Both visibility values are larger than the threshold of 70.7%, which is the benchmark for Bell's inequality, demonstrating that the two memories are truly entangled in these hybrid DOFs.

We also use an entanglement witness to characterize whether the two memories exhibit hybrid entanglement. The witness is expressed as ref. 37

$$W = V_1 + V_2 \leq 1 \qquad (7)$$

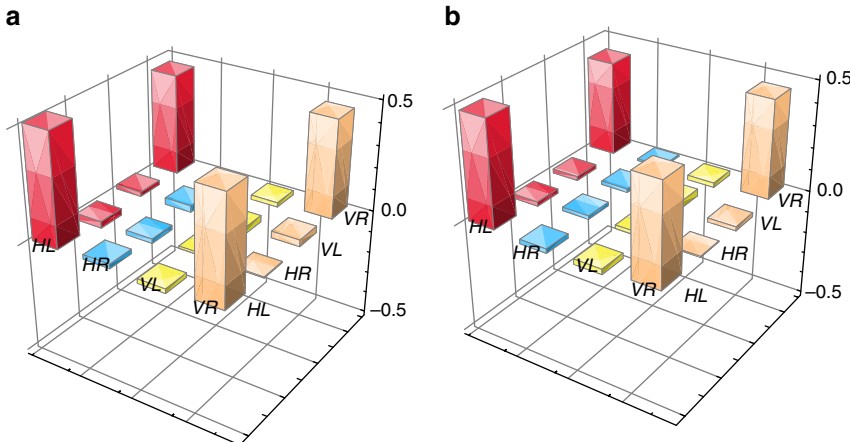

**Figure 4 | Density matrices for hybrid entanglement.** The real parts of the constructed density matrices for hybrid entanglement before (**a**) and after (**b**) storage. The values of the imaginary parts are very low; they are distributed within ranges of $0 \pm 2.9\%$ before storage and $0 \pm 3.8\%$ after storage.

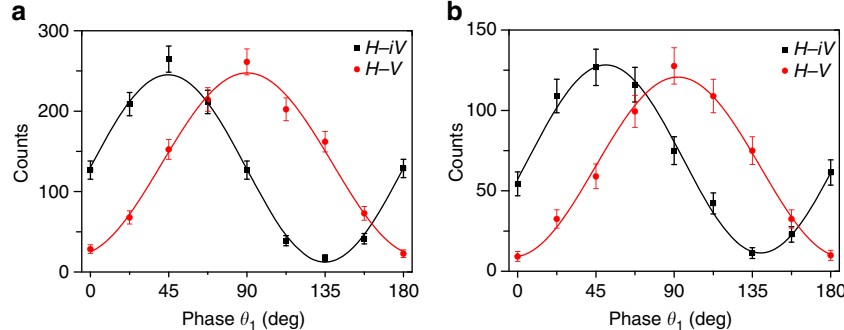

**Figure 5 | Interference curves for hybrid entanglement.** The interference of the two-photon correlations before (**a**) and after (**b**) storage. The error bars are estimated from Poisson statistics and represent $\pm$ s.d. All the data are raw and not subjected to noise correction.

where $V_1$ and $V_2$ are the visibilities of two-photon interference in the diagonal/anti-diagonal and left-circular/right-circular bases, respectively, of the Signal 2 photon. If there is no entanglement, then the measured witness should not be violated, which would mean that the state is separable. Our experimental results indicate that $W = V_1 + V_2 = 1.861 \pm 0.007 \gg 1$ before storage and $W = V_1 + V_2 = 1.691 \pm 0.025 \gg 1$ after storage, hinting that there must be hybrid entanglement between our separated memories.

## Discussion

We experimentally realize the quantum storage of entanglement in multiple DOFs based on cold atomic systems, thereby establishing multi-DOF memory–memory entanglement (see Supplementary Fig. 4 and Supplementary Note 5). This entanglement, including both hyperentanglement and hybrid entanglement, is verified by constructing density matrices and by checking the CHSH inequality or an entanglement witness. We believe that our experiment is useful for research on atom-based quantum networks.

Recently, the quantum teleportation of multiple DOFs of a single photon has been demonstrated[38]. By combining multi-DOF quantum storage with multi-DOF teleportation, a multi-DOF quantum repeater[39] with a high communication capacity is expected to be achieved. Moreover, with an increasing number of DOFs and an increasing number of dimensions in a single DOF, for example, using the time-frequency DOF[40] and

higher dimensions of the OAM DOF[23], entanglement in a considerably expanded Hilbert space can be established between atomic ensembles, thereby providing a versatile platform for demonstrating quantum computing protocols and complex quantum networks.

## Methods

**Experimental details for the systems.** The system operates periodically with a cycle time of 10 ms, which includes 8.6 ms for atomic trapping and initial state preparation and 1.4 ms of operation time, consisting of 2,800 cycles with a cycle time of 500 ns. In each cycle, the Pump 1, Pump 2 and coupling light beams are pulsed by an acousto-optic modulator. All of them are Gaussian-shaped beams with a waist of 2.2 mm. The optical depths of the atomic ensembles in MOT A and MOT B are $\sim 20$ and 50, respectively.

**Density matrix reconstruction.** For the 2D polarization entanglement, individual projection measurements related to the projection of the entangled state into the four bases $|H\rangle$, $|V\rangle$, $(|H\rangle - i|V\rangle)^{1/2}$, and $(|H\rangle + |V\rangle)^{1/2}$ are performed on each photon before and after the storage of the Signal 1 photon with the shutdown of SLM 1 and SLM 2.

For the 3D OAM entanglement, we project the Signal 1 and Signal 2 photons onto SLM 1 and SLM 2 with nine different phase states $|\psi_{1-9}\rangle$, corresponding to the states $|L\rangle$, $|G\rangle$, $|R\rangle$, $(|G\rangle + |L\rangle)/2^{1/2}$, $(|G\rangle + |R\rangle)/2^{1/2}$, $(|G\rangle + i|L\rangle)/2^{1/2}$, $(|G\rangle - i|R\rangle)/2^{1/2}$, $(|L\rangle + |R\rangle)/2^{1/2}$, and $(|L\rangle + i|R\rangle)/2^{1/2}$.

For the hybrid entanglement, the density matrices of the $|\psi'_{1\text{-hybrid}}\rangle$ and $|\psi'_{2\text{-hybrid}}\rangle$ states are reconstructed by projecting Signal 2 into the four polarization bases of $|H\rangle$, $|V\rangle$, $(|H\rangle - i|V\rangle)/2^{1/2}$ and $(|H\rangle + |V\rangle)/2^{1/2}$ and Signal 1 into the four OAM bases of $|L\rangle$, $|R\rangle$, $(|L\rangle - i|R\rangle)/2^{1/2}$ and $(|L\rangle + |R\rangle)/2^{1/2}$.

**Data availability.** The data that support the findings of this study are available from the corresponding authors upon reasonable request.

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

## Acknowledgements

This work was supported by the National Fundamental Research Program of China (Grant No. 2 011CBA00200) and the National Natural Science Foundation of China (Grant Nos. 11174271, 61275115, 61435011 and 61525504).

## Author contributions

D.-S.D. conceived the experiment. The experimental work and data analysis were conducted by W.Z. and D.-S.D. with assistance from, M.-X.D., S.S., K.W., S.-L.L., Y.L., and Z.-Y.Z. Moreover, W.Z. and D.-S.D. wrote this paper with assistance from B.-S.S., and B.-S.S. and G.-C.G. supervised the project.

## Additional information

**Competing financial interests:** The authors declare no competing financial interests.

**Publisher's note**: 

