## [Peer Review File · Nature Communications]

Reviewers' comments:

Reviewer #1 (Remarks to the Author):

The authors report that they have demonstrated hyper- and hybrid-entanglement between spatially separated atomic ensembles using multiple degrees of freedom. The manuscript contains convincing evidence in support of this claim. In my opinion, this is a significant result and worthy of publication in a high-profile journal.

To date, there have been few demonstrations of memory-to-memory entanglement and only one demonstration of storage of hyper-entanglement. This manuscript is therefore the first to report hyper-entanglement between two memories, the first to show hyper-entanglement storage in a cold-atom system and the first to use the spatial mode of the atomic coherence as the hyper-entangled degree of freedom. The manuscript further demonstrates that with only a minor modification, it can be used to demonstrate hybrid-entanglement as well.

While the building blocks of the present experiment have been previously reported in literature, for example the storage and retrieval of orbital angular momentum entanglement [32] (references as numbered in the manuscript) and the scheme for producing hyper-entanglement [10], the combination of these elements into a functional memory represents a significant advance.

Overall, I found the manuscript well-written and easy to follow. (Numerous grammatical errors will need to be corrected, but these do not affect the clarity of the manuscript). The section motivating the use of hyper-entanglement also provided a good review of existing literature. I do, however, have some questions and comments that I would like the authors to consider.

1) In the first paragraph, the phrase "...we experimentally established hyper- and hybrid-entanglement in multiple degrees of freedom including path, polarization, and orbital angular momentum between two separated atomic ensemble..." could be interpreted to suggest that three degrees of freedom were entangled, when in fact it was only two. While polarisation is used to communicate the entangled state between ensembles, only spatial degrees of freedom of the atomic coherence (k -vector and angular momentum number) are involved in the memory-to-memory hyper-entanglement. This is clear later in the text, but care should be taken with the wording in the first paragraph.

2) Certain quantities relating to the memory performance and other efficiencies are contained in the supplementary materials. It would be useful if the main text referenced these so the interested reader could find them easily.

3) What is the probability of generating entanglement on each 500 ns run of the experiment (corrected for loss)? Further, what is the probability of detecting a successful event assuming that an entangled pair is produced (not corrected for loss)? What is the overall rate of successful events? This would be particularly useful to quantify since the authors

state that a low count rate is one of the reasons for not verifying that entanglement between different degrees of freedom is independent.

4) How does the event rate compare to the dark-count rate of the photo-detectors?

5) How many events are used to compute the density matrices and fidelities that are reported?

6) Including a legend on Fig.1 (b) for the various optical components would make it easier to read (it took me a little while to find the $\lambda/4$ label).

7) I'm a bit puzzled by figure S4 in the supplementary material. Can the efficiency of 25% be inferred from the figure, or is it measured separately? Also, I would expect that the pulses would have a two-sided exponential decay shape. Is there a reason that they are Gaussian?

Reviewer #2 (Remarks to the Author):

In their manuscript, Zhang et al. report on entanglement between 2 memories made with cold atoms. While this has been shown in various systems already, the authors report on matter-matter entanglement in multiple degrees of freedom. They claim that this might find interesting applications for efficient quantum communication using both high channel capacity and efficient information processing. They start by creating photon-spin wave entanglement using polarisation encoding and orbital angular momentum from spontaneous Raman scattering. They subsequently map the photonic properties to spin waves in another atomic ensemble, which results in spin-wave spin-wave hyper-entanglement. To reveal entanglement between the 2 atomic ensembles, they map the state of spin-waves to photons and perform several measurements at the photonic level. With minor modifications of the setup, they create and detect hybrid entangled states in path and orbital angular momentum. While I am a priori positive with the proposed results, I would like the authors answer the following questions before I can recommend publication in Nature Communications.

1-The authors use several paragraphs in the introduction to describe the potential of entanglement in multiple degrees of freedom. In particular, they claim that hyper-entanglement could greatly enhance the distribution rates in long-distance quantum communication based quantum repeaters as hyper-entanglement is known to allow for deterministic Bell state discrimination. This is controversial as the author forget to say that once a swapping is realized deterministically using hyper-entanglement, the resulting state is no longer hyper entangled but is a standard 2 qubit entanglement. Therefore, hyper entanglement only helps at the first level of the swapping operations in quantum repeaters and not for all the levels. This allows one to improve the distribution rate by a factor of 2 essentially and given the complexity of generating, storing and detecting hyper-entangled states, I would say there is essentially no-advantage of using hyper entanglement for quantum repeaters. I thus suggest that either the authors better justify their claim or that

they remove the corresponding paragraph. I am personally in favor of the second choice as the section regarding motivation is much too long.

2-Once the introduction is made shorter, the authors can provide more information about the way entanglement is created. For example, when they say "Signal-1 single photons at 795-nm wavelength hyperentangled with atomic spin wave in MOT A is created with the aid of beam displacer (BD) after the illumination of Pump-1 light, and then is delivered to the second atomic ensemble in MOT B for storage." This is not clear how a beam displacer can create hyperentanglement. More generally, the authors do not give information about the way hyper and hybrid entanglement are created. A lot of details are done regarding the detections but there is not enough information to understand how photon-spin wave entanglement is created.

3-While their optical depths are high, the efficiencies of the read out of the first memory and the storage and readout in the second one are not impressively high. Can the authors comment on the limitations regarding the memory efficiencies?

4-As for the title, I am not sure that the present form does provide a good summary of what is done in the manuscript. In particular, entanglement can be stored without creating matter-matter entanglement. Can the author propose a better title?

Reviewer #3 (Remarks to the Author):

The authors have presented an interesting demonstration of "memory to memory" quantum state transfer. In the process of transferring the quantum state, they "swap" the entanglement from polarization to Orbital Angular Momentum entanglement, which seems similar to the methods used by Anton Zeilingers' team.

To my knowledge, I haven't seen such a demonstration previously and the work is very complete in presentation, with a lot of data that seems to confirm the preservation of the entanglement in the transfer between memory MOTs. The couple of parts that seem a bit a miss for me is how this really creates a step change in development of a quantum repeater, and the the use of "hybrid entanglement" seems out of place to me.

1. Step Change in the Development of Quantum Repeater

The author have a clear justification in their introduction that their work is presenting a substantive contribution towards the development of quantum repeaters. Such repeaters are essential for future Quantum Networks and Computers. However, I find it hard to see how this is an example of a such a devices. The amplification arising from the storage in the MOT is not a new concept, with numerous examples including several already published by the senior author, and the work by B.C. Buchler's team in Australia.

The demonstration seems to an example of entanglement conversion between to memory "banks". That in it's self could be a useful feature. I think the authors need to more clearly justify why this particular example of Photon storage and re-emission is directly resulting in

a new achievement in quantum repeater design.

2. Hybrid Entanglement

The author repeatedly use the term hybrid entanglement, and I find it a little confusing. As far as i understand, Hybrid entanglement is considered the entanglement between a "classical" arm and a "quantum" arm that has many benefits as comms systems move towards the quantum regime (Nature Photonics 8,564-569(2014)). It would seem to be that no such demonstration is given in the paper. I do however wonder if the authors are meaning to refer to Hyper Entanglement (Nature Communications 6,174,2011), where Paul Kwiat's team have demonstrated example of of super dense coding. I feel the authors should offer commentary on there use of Hybrid Entanglement.

With suitable clarification, and justification of this works impact on the generation of novel quantum repeaters I feel this work could be suitable for publication in Nature Communication.

The responses to the comments given by Reviewer #1

We thank the Reviewer #1 very much for praising our work as “a significant result and worthy of publication in a high-profile journal” and giving us very positive comments. We have considered other comments very carefully and revised the manuscript completely along these comments. The main revisions and the detailed responses made are as follows:

Reviewer #1 (Remarks to the Author):

The authors report that they have demonstrated hyper- and hybrid-entanglement between spatially separated atomic ensembles using multiple degrees of freedom. The manuscript contains convincing evidence in support of this claim. In my opinion, this is a significant result and worthy of publication in a high-profile journal.

To date, there have been few demonstrations of memory-to-memory entanglement and only one demonstration of storage of hyper-entanglement. This manuscript is therefore the first to report hyper-entanglement between two memories, the first to show hyper-entanglement storage in a cold-atom system and the first to use the spatial mode of the atomic coherence as the hyper-entangled degree of freedom. The manuscript further demonstrates that with only a minor modification, it can be used to demonstrate hybrid-entanglement as well.

While the building blocks of the present experiment have been previously reported in literature, for example the storage and retrieval of orbital angular momentum entanglement [32] (references as numbered in the manuscript) and the scheme for producing hyper-entanglement [10], the combination of these elements into a functional memory represents a significant advance.

Overall, I found the manuscript well-written and easy to follow. (Numerous grammatical errors will need to be corrected, but these do not affect the clarity of the manuscript). The section motivating the use of hyper-entanglement also provided a good review of existing literature. I do, however, have some questions and comments that I would like the authors to consider.

Reply:

We have corrected the grammatical errors and improved the English significantly.

1) In the first paragraph, the phrase "...we experimentally established hyper- and hybrid-entanglement in multiple degrees of freedom including path, polarization, and orbital angular momentum between two separated atomic ensemble..." could be interpreted to suggest that three degrees of freedom were entangled, when in fact it was only two. While polarisation is used to communicate the entangled state between ensembles, only spatial degrees of freedom of the atomic coherence (k-vector and angular momentum number) are involved in the memory-to-memory hyper-entanglement. This is clear later in the text, but care should be taken with the wording in the first paragraph.

Reply:

We are very sorry for our misleading words. According to this comment, we deleted the word “polarization”. After that, the corresponding paragraph is changed to “we experimentally established hyper- and hybrid- entanglement in multiple degrees of freedom including path

(*K*-vector) and orbital angular momentum between two separated atomic ensembles by using quantum storage”. Further for clarity, we added an explanation in the paragraph above the Fig. 1--- “(Hereafter, we use “path” to represent corresponding “*K*-vector” through the whole text)”.

2) Certain quantities relating to the memory performance and other efficiencies are contained in the supplementary materials. It would be useful if the main text referenced these so the interested reader could find them easily.

Reply:

In the second paragraph of Discussion, we added “See Supplementary material for memory performance and efficiencies”.

3) What is the probability of generating entanglement on each 500 ns run of the experiment (corrected for loss)? Further, what is the probability of detecting a successful event assuming that an entangled pair is produced (not corrected for loss)? What is the overall rate of successful events? This would be particularly useful to quantify since the authors state that a low count rate is one of the reasons for not verifying that entanglement between different degrees of freedom is independent.

Reply:

Experimentally, we obtain ~500 coincidence counts in total 200s for the retrieved photons, corresponding to ~2.5 counts per second. So, on each 500 ns run, the probability (corrected for loss) of generating entanglement between two atomic ensembles is estimated as $2.5/(2800 \times 100 \times 75\% \times 65\% \times 45\% \times 50\% \times 50\% \times 75\% \times 75\%) \sim 2.9 \times 10^{-4}$, in which 2800×100 is the trial number per second, the first 75% is coupling efficiency for signal 1 from space to fibre, $65\% \times 45\%$ is for cavity transmission efficiency, $50\% \times 50\%$ is the total detecting efficiencies of two detectors, $75\% \times 75\%$ is the total reflectivity of two SLMs.

Assuming an entangled pair is produced, the probability of detecting a successful event is $1 \times 75\% \times 65\% \times 45\% \times 50\% \times 50\% \times 75\% \times 75\% \sim 3.1 \times 10^{-2}$.

The overall rate of successful events is 2.5/s accounting for ~500 coincidence counts in total 200s.

Due to the low count rate, it always takes a few hundred seconds to get one data. If taking the method of joint measurement including multiple DOFs, we have to take at least 100 hours to fill the 36×36 data matrix for reconstructing density matrix. So we use methods as illustrated in the Supplementary material to verify the independence between two degrees of freedom. For clear description, we add the related probability in Supplementary material.

4) How does the event rate compare to the dark-count rate of the photo-detectors?

Reply:

During our experiment, the dark-count rate is ~180/s for both detectors while the photon count rate (including dark-count rate) is ~800/s. In Fig S4, we can see that the Signal to Noise Ratio (SNR) of the retrieved signal is ~25 (through statistical calculation), which means the overall successful event rate degrades from 2.5/s to $2.5 \times (25-1)/25 = 2.4/s$ due to accidental coincidences coming from dark-count or environmental noise. So the effect of dark count and

environmental noise is rather limited. We added these rates in Supplementary material.

5) How many events are used to compute the density matrices and fidelities that are reported?

Reply:

Before storage in MOT B, the counts are accumulated in 100s, the maximum counts (for photons projecting bases H-H or V-V or LL RR GG) is ~900. After storage in MOT B, the counts which are accumulated in 200 s, are ~500.

6) Including a legend on Fig.1 (b) for the various optical components would make it easier to read (it took me a little while to find the $\lambda/4$ label).

Reply:

We included a legend on Fig. 1(b) according to this comments.

7) I'm a bit puzzled by figure S4 in the supplementary material. Can the efficiency of 25% be inferred from the figure, or is it measured separately? Also, I would expect that the pulses would have a two-sided exponential decay shape. Is there a reason that they are Gaussian?

Reply:

*We are very sorry for our inadequate explanation of Fig S4. Actually, the first pulse in Fig S4 (in old version) is the leaked signal, and the second pulse is the retrieved signal. So the efficiency of 25% is calculated by the ratio of **retrieved pulse** to **pulse without storage**. The Fig S4 (in old version) is to present the SNR of the coincidence (which is ~25) for stored signal. Now, we update Fig S4 in a new version in which signal without storage in MOT B is also included, thus the efficiency can be estimated accordingly.*

We completely agree with the reviewer that pulses in Fig. S4 have a two-sided exponential decay shape. We make a presentation error here: the real meaning we indeed want to express is “the 100-ns memory efficiency in MOT B is ~25% for photon carrying no OAM information as depicted in Fig. S4 and is ~20% for photon carrying OAM ...”, we do not mean that the pulse has the Gaussian shape in old version. We are very sorry for this misleading expression. We have revised this sentence.

The responses to the comments given by Reviewer #2

We thank the Reviewer #2 very much for her/his “p priori positive” attitude toward our work and giving us helpful comments to improve our work. We have considered these comments very carefully and revised the manuscript completely along these comments. The main revisions and the detailed responses made are as follows:

Reviewer #2 (Remarks to the Author):

In their manuscript, Zhang et al. report on entanglement between 2 memories made with cold atoms. While this has been shown in various systems already, the authors report on matter-matter entanglement in multiple degrees of freedom. They claim that this might find interesting applications for efficient quantum communication using both high channel capacity and efficient information processing. They start by creating photon-spin wave entanglement using polarisation

encoding and orbital angular momentum from spontaneous Raman scattering. They subsequently map the photonic properties to spin waves in another atomic ensemble, which results in spin-wave hyper-entanglement. To reveal entanglement between the 2 atomic ensembles, they map the state of spin-waves to photons and perform several measurements at the photonic level. With minor modifications of the setup, they create and detect hybrid entangled states in path and orbital angular momentum. While I am a priori positive with the proposed results, I would like the authors answer the following questions before I can recommend publication in Nature Communications.

1-The authors use several paragraphs in the introduction to describe the potential of entanglement in multiple degrees of freedom. In particular, they claim that hyper-entanglement could greatly enhance the distribution rates in long-distance quantum communication based quantum repeaters as hyper-entanglement is known to allow for deterministic Bell state discrimination. This is controversial as the author forget to say that once a swapping is realized deterministically using hyper-entanglement, the resulting state is no longer hyper entangled but is a standard 2 qubit entanglement. Therefore, hyper entanglement only helps at the first level of the swapping operations in quantum repeaters and not for all the levels. This allows one to improve the distribution rate by a factor of 2 essentially and given the complexity of generating, storing and detecting hyper-entangled states, I would say there is essentially no-advantage of using hyper entanglement for quantum repeaters. I thus suggest that either the authors better justify their claim or that they remove the corresponding paragraph. I am personally in favor of the second choice as the section regarding motivation is much too long.

Reply:

We thank the reviewer 2 for pointing out this controversial view on the advantages of enhancing distribution efficiency for quantum repeater by using hyper-entanglement. Following the suggestion of the reviewer, we have removed the corresponding paragraph in the introduction part.

2-Once the introduction is made shorter, the authors can provide more information about the way entanglement is created. For example, when they say "Signal-1 single photons at 795-nm wavelength hyper-entangled with atomic spin wave in MOT A is created with the aid of beam displacer (BD) after the illumination of Pump-1 light, and then is delivered to the second atomic ensemble in MOT B for storage." This is not clear how a beam displacer can create hyper-entanglement. More generally, the authors do not give information about the way hyper and hybrid entanglement are created. A lot of details are done regarding the detections but there is not enough information to understand how photon-spin wave entanglement is created.

Reply:

We are very sorry for our unclear presentation. We have added more sentences and words to describe the generation of the hyper-entanglement and the hybrid entanglement between photon and spin wave.

The revised sentences are below "Signal-1 single photon at 795-nm wavelength hyperentangled with the spin wave in MOT A is created with the aid of a beam displacer (BD) after the illumination of Pump-1 light. Due to the conservation of the momentum in the SRS process, the initial system has zero linear momentum and OAM, thus the resulting joint state of

Signal 1 and the spin wave has zero momentum both in K -vector space and OAM space, so the spin wave in MOT A entangles with Signal-1 photon. BD here is used for coherently superposing the state of Signal 1 from different K -vector space to the same path. This hyperentangled state is expressed as Eq. 1 (Hereafter, we use the “path” to represent corresponding “ K -vector” through the whole text). The generated Signal 1 photon is then delivered to the second atomic ensemble in MOT B for storage.”

For generating a hybrid entanglement, the preparation is quite the same, we revised the corresponding paragraph as “With a little change in the experimental setup, we generate polarization-path entanglement between Signal-1 photon and the spin wave in MOT A, which utilizes the principle of momentum conservation in K -vector space.” Now, we believe the manuscript give a more clear and easy-understanding presentation.

3-While their optical depths are high, the efficiencies of the read out of the first memory and the storage and readout in the second one are not impressively high. Can the authors comment on the limitations regarding the memory efficiencies?

Reply:

We thank the reviewer very much for his/her constructive comments.

Actually, a large OD~150 can reach a high memory efficiency of ~78% (~70%) in a backward (forward) setup in a cold atomic ensemble for a weak coherent light pulse [Phys. Rev. Lett. 110 083601(2013)]. Further increasing OD will lead a near-unity storage efficiency [arXiv: 1605.08519] for weak coherent light. However, for a true single photon, the maximum efficiency is about 50% at present even with the optimization of the pulse shape in a forward setup [Optics Express 20, 24124-24131 (2012)]. In our experiment, the OD is of ~50 and there is no modulation of the pulse shape. Besides, the resident external magnetic field, which leads to a rapid decay of the spin wave, also decreases the efficiency. Overall, the efficiency achieved in our experiment is only 25%. The efficiency can be improved by optimizing the pulse shape, increasing the OD, reducing the resident external magnetic field.

4-As for the title, I am not sure that the present form does provide a good summary of what is done in the manuscript. In particular, entanglement can be stored without creating matter-matter entanglement. Can the author propose a better title?

Reply: *We agree with the Reviewer’s comment. So we changed the title to “Experimental Realization of Memory-Memory Entanglement in Multiple Degrees of Freedom”.*

The responses to the comments given by Reviewer #3

We thank the Reviewer #3 very much for her/his positive attitude toward our work and giving us helpful comments to improve our work. We have considered these comments very carefully and revised the manuscript completely along these comments. The main revisions and the detailed responses made are as follows:

Reviewer #3 (Remarks to the Author):

The authors have presented an interesting demonstration of "memory to memory" quantum state

transfer. In the process of transferring the quantum state, they "swap" the entanglement from polarization to Orbital Angular Momentum entanglement, which seems similar to the methods used by Anton Zeilingers' team.

To my knowledge, I haven't seen such a demonstration previously and the work is very complete in presentation, with a lot of data that seems to confirm the preservation of the entanglement in the transfer between memory MOTs. The couple of parts that seem a bit a miss for me is how this really creates a step change in development of a quantum repeater, and the use of "hybrid entanglement" seems out of place to me.

Reply:

We are very sorry for unclear descriptions in introduction part. As this comments are similar to two questions listed below, so we answer these questions point by point accordingly in the below parts.

1. Step Change in the Development of Quantum Repeater

The author have a clear justification in their introduction that their work is presenting a substantive contribution towards the development of quantum repeaters. Such repeaters are essential for future Quantum Networks and Computers. However, I find it hard to see how this is an example of a such a devices. The amplification arising from the storage in the MOT is not a new concept, with numerous examples including several already published by the senior author, and the work by B.C. Buchler's team in Australia

The demonstration seems to an example of entanglement conversion between to memory "banks". That in it's self could be a useful feature. I think the authors need to more clearly justify why this particular example of Photon storage and re-emission is directly resulting in a new achievement in quantum repeater design.

Reply:

We are very sorry for over emphasizing on the advantages of enhancing distribution efficiency for quantum repeater by using hyper-entanglement. We have revised the introduction, removed the corresponding paragraph in the introduction part.

As pointed out by reviewer # 2, using hyper-entanglement may allow one improve the distribution rate by a fact of 2, therefore there is no big advantage compared with a standard entanglement in one DOF. The reviewer # 2 suggests us to remove the corresponding paragraph about quantum repeater. After we considered this comment carefully, we agreed with the reviewer # 2 on this comment, and removed the corresponding paragraph accordingly. We are sorry again for over emphasizing on the advantages by using hyper-entanglement in quantum repeater.

2. Hybrid Entanglement

The author repeatedly use the term hybrid entanglement, and I find it a little confusing. As far as i understand, Hybrid entanglement is considered the entanglement between a "classical" arm and a "quantum" arm that has many benefits as comms systems move towards the quantum regime (Nature Photonics 8,564-569(2014)). It would seem to be that no such demonstration is given in the paper. I do however wonder if the authors are meaning to refer to Hyper Entanglement (Nature

Communications 6,174,2011), where Paul Kwiat's team have demonstrated example of of super dense coding. I feel the authors should offer commentary on there use of Hybrid Entanglement.

Reply:

The hybrid entanglement we referred in our text is a nonseparable superposition between two different DOFs (e.g. one for polarization DOF, the other for OAM DOF), as some other groups define in their works [for example, Opt. Express 18. 18243 (2010), Ref. 17 in our new text], we call it hybrid entanglement.

We are sorry for being unable to find the wok (Nature Communications 6,174, (2011)). The reviewer may refer to the work in Nature Communications 6, 7185 (2015). In this work, they have same definition with our work, in which the entanglement in multiple DOFs is called as hyper-entanglement. We have cited this work from Kwiat's group as Ref. 16.

With suitable clarification, and justification of this works impact on the generation of novel quantum repeaters I feel this work could be suitable for publication in Nature Communication.

REVIEWERS' COMMENTS:

Reviewer #1 (Remarks to the Author):

After reviewing the modifications made to the manuscript, I am satisfied that the authors have sufficiently addressed the points that I raised during the first round of review. In my opinion, the manuscript has been improved in response to the comments by all three referees and I am pleased to recommend it for publication in Nature Communications.

I will, however, also weigh in on a point raised by referee #3: the use of the term hybrid-entanglement. The authors' use of the term is consistent with a number of publications that use it to describe entanglement between different degrees of freedom. Some recent publications use it to describe a particular case of hybrid-entanglement: when one of the degrees of freedom involves a "large" quantum state (eg. a coherent state).

I think that the authors are well-justified in their use of the term, but the phrase that introduces it in the manuscript could be more clear. Currently it reads: "Being the same with the hyperentanglement, hybrid entanglement is another important state taking advantages of different DOFs." Given that the term has two different usages, the authors should explicitly say what they mean by hybrid-entanglement and how it differs from hyperentanglement.

Reviewer #2 provided confidential remarks to the editor, supporting publication.

Reviewer #3 (Remarks to the Author):

I feel the authors have adequately addressed my concerns, and I support its publication in Nature Communications.

The responses to the comments given by Reviewer #1

We thank the Reviewer #1 very much for recommending our work for publication in Nature Communications and giving us very useful comments. We revised the manuscript accordingly along these comments. The main revisions and the detailed responses made are as follows:

Reviewer #1 (Remarks to the Author):

After reviewing the modifications made to the manuscript, I am satisfied that the authors have sufficiently addressed the points that I raised during the first round of review. In my opinion, the manuscript has been improved in response to the comments by all three referees and I am pleased to recommend it for publication in Nature Communications.

I will, however, also weigh in on a point raised by referee #3: the use of the term hybrid-entanglement. The authors' use of the term is consistent with a number of publications that use it to describe entanglement between different degrees of freedom. Some recent publications use it to describe a particular case of hybrid-entanglement: when one of the degrees of freedom involves a "large" quantum state (eg. a coherent state).

I think that the authors are well-justified in their use of the term, but the phrase that introduces it in the manuscript could be more clear. Currently it reads: "Being the same with the hyperentanglement, hybrid entanglement is another important state taking advantages of different DOFs." Given that the term has two different usages, the authors should explicitly say what they mean by hybrid-entanglement and how it differs from hyper entanglement..

Reply:

We revised the corresponding sentences to be "Another important type of state that takes advantage of multiple DOFs is hybrid entanglement. Whereas hyperentanglement is the entangling of states in multiple DOFs independently, hybrid entanglement is the entangling of states in multiple DOFs mutually, which has a similar form to that discussed in Ref. 17."